# Computational Imaging for Simultaneous Image Restoration and Super-Resolution Image Reconstruction of Single-Lens Diffractive Optical System

**Kai Liu [1,2], Xiao Yu [1], Yongsen Xu [1], Yulei Xu [1,*], Yuan Yao [1], Nan Di [3], Yefei Wang [1], Hao Wang [1] and Honghai Shen [1]**

1   Key Laboratory of Airborne Optical Imaging and Measurement, Changchun Institute of Optics, Fine Mechanics and Physics, Chinese Academy of Sciences, Changchun 130033, China; liukai174@mails.ucas.edu.cn (K.L.); m13324473070@163.com (X.Y.); pm13l@sina.com (Y.X.); c8george@sina.com (Y.Y.); rebecca2946@163.com (Y.W.); wanghao7600@163.com (H.W.); shenhh@ciomp.ac.cn (H.S.)
2   University of Chinese Academy of Sciences, Beijing 100049, China
3   Academy for Advanced Interdisciplinary Studies, Northeast Normal University, Changchun 130024, China; din@ciomp.ac.cn
*   Correspondence: yuleixu@ciomp.ac.cn; Tel.: +86-138-4406-5873

**Abstract:** Diffractive optical elements (DOEs) are difficult to apply in natural scenes imaging covering the visible bandwidth-spectral due to their strong chromatic aberration and the decrease in diffraction efficiency. Advances in computational imaging make it possible. In this paper, the image quality degradation model of DOE in bandwidth-spectral imaging is established to quantitatively analyze its degradation process. We design a DDZMR network for a single-lens diffractive lens computational imaging system, which can simultaneously perform image restoration and image super-resolution reconstruction on degraded images. The multimodal loss function was created to evaluate the reconstruction of the diffraction imaging degradation by the DDZMR network. The prototype physical prototype of the single-lens harmonic diffraction computational imaging system (SHDCIS) was built to verify the imaging performance. SHDCIS testing showed that optical chromatic aberration is corrected by computational reconstruction, and the computational imaging module can interpret an image and restore it at 1.4 times the resolution. We also evaluated the performance of the DDZMR model using the B100 and Urban100 datasets. Mean Peak Signal to Noise Ratio (PSNR)/Structural Similarity (SSIM) were, respectively, 32.09/0.8975 and 31.82/0.9247, which indicates that DDZMR performed comparably to the state-of-the-art (SOTA) methods. This work can promote the development and application of diffractive imaging systems in the imaging of natural scenes in the bandwidth-spectrum.

**Keywords:** harmonic diffractive optical element; computational imaging; image restoration; image super-resolution

## 1. Introduction

The most important advantages of planar DOE are their light weight and simple structure, which allow for the simplified design of optical systems. DOE can also be transplanted to film substrates to provide large-aperture thin-film optical imaging, which is valuable for the development of large-aperture high-resolution optical imaging payloads [1]. Broad bandwidth spectral imaging is difficult to implement in modern optical sensor systems using single-lens DOE [2]. The two main reasons for this are that the diffraction efficiency for all light waves deviating from the design center wavelength drops sharply [3,4], and strong chromatic aberration that varies with wavelength [5–7]. Chromatic aberration is caused by the dispersion of light waves of different wavelengths on the surface of a medium. Chromatic aberration is eliminated in the conventional solution by placing

the an inverse power DOE with the same dispersion in the aberration correcting lens group [1]. However, this complicates the system for and makes it difficult for widespread low-cost applications. Emerging computational imaging techniques enables replace complex lens group for the task of correcting aberrations, which encourages the development of single-lens DOE imaging. Accordingly, researchers are increasingly interested in the field of simple DOE imaging applying computational imaging techniques [2–4,6–12].

Harmonic diffractive optical elements (HDOE) have more unique performance than ordinary DOEs in bandwidth spectral imaging. When the wavelength of the light wave is equal to the central wavelength or the harmonic wavelength, the diffraction efficiency of the harmonic diffractive optical element can reach 100% [10]. From this point of view, it is more advantageous to apply in broadband spectral imaging.

Chromatic aberration is inevitable for single-lens DOE bandwidth imaging systems [8]. It is, therefore, necessary to combine computational imaging reconstruction algorithms to suppress chromatic aberration and improve image quality [11]. Traditional deconvolution methods are not robust in using natural image priors when dealing with the challenging large blur kernels of diffractive systems [13,14]. Algorithms based on data-driven self-similarity show more promising image restoration or reconstruction performance [4,6–8,10,15]. Cross-channel priors combined with a feed-forward deep convolutional neural network deconvolution algorithm for reconstructing images from diffraction imaging can effectively eliminate artifacts and blur caused by chromatic aberration [10]. Although the above algorithm can restore and reconstruct diffraction imaging to eliminate the influence of chromatic aberration, it is still not robust enough for single-lens diffraction wide-spectrum imaging. The recovery of image details and the removal of stray light noise need to be improved.

Image super-resolution (ISR) processing algorithms can increase the resolution of an imaging system without upgrading the imaging system hardware. Fusion of the single-frame image ISR algorithm into the computational imaging module can improve imaging systems' performance. Deep learning models have become more widely used in ISR algorithms because they offer significant improvements over conventional algorithms [16–23]. Recent ISR Networks employ a residual learning framework [17,18] to learn high-frequency image details, which are then added to the input low resolution(LR) images to produce super-resolution images. Data-driven deep learning network model ISR algorithms vary in their architectural design. Other network techniques that produce ISR images include compressed sensing [19], densely connected networks [20], Laplacian deep convolutional networks [21], multiple attention mechanisms [22], and generative adversarial networks (GAN) [23].

In this study, we analyzed HDOE focusing performance in wide-bandwidth spectral applications. The imaging degradation caused by chromatic aberration and diffraction efficiency reduction in HODE wide-spectrum imaging is analyzed. We described improvements made to the computational imaging module by combining the image restoration algorithm with the image super-resolution algorithm. We design a DDZMR network for simultaneous image restoration and image super-resolution, and customize a multimodal objective function based on the HDOE image quality degradation model. We also built a demonstration prototype of the application of a SHDCIS in the visible spectrum, as shown in Figure 1. The results of the experiments show that the imaging capability of the prototype can reach the level of photographic imagery and it produced satisfactory results for deblurring, correcting chromatic aberration and ISR.

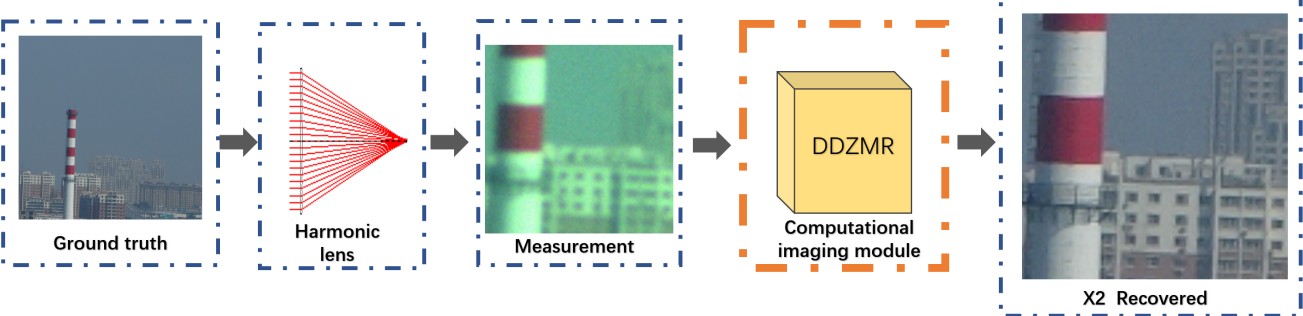

**Figure 1.** Schematic diagram of SHCIS: degraded low-resolution images are measured by single-lens HDOE and color RGB detectors.The DDZMR network performs image restoration and image super-resolution on degraded images to generate high-quality images.

## 2. Analysis of Imaging Properties of HDOE in Wide-Bandwidth Spectrum

The focus blur of visible spectrum bandwith is a long-standing problem in diffractive lens [2]. The research on the imaging characteristics of HDOE in the bandwidth spectrum and the quantitative analysis of its image quality degradation are of great benefit to the construction of its image post-processing algorith. In this section, the spectral characteristics of HDOE were firstly analyzed, and then the image quality degradation model of HDOE under bandwidth spectrum was constructed.

### 2.1. Analysis of Spectral Characteristics of HDOE

HDOE has the same optical power at multiple discrete harmonic wavelengths and can achieve 100% diffraction efficiency, which is more advantageous than ordinary DOE in imaging in the bandwidth spectrum. Then, the HDOE was designed using commercial optical design software to achieve diffractive imaging in the full spectral range of visible light. We set the design center wavelength to the median of the visible light spectrum ($\lambda_0$ = 555 nm), to which the human eye is most sensitive, and set the diffraction order of the center wavelength to the 10th order ($m = 10$). The purpose of this is to generate enough harmonics to cover the visible spectrum. The phase function of the HDOE is optimized as a continuous and increasing higher-order term function in commercial optical design software, which can be expressed as:

$$\varphi(r) = m \sum_{i=1}^{N} A_i r^{2i},\tag{1}$$

where $m$ is a constant that represents the diffraction design order of the center wavelength, $N$ is the order of the phase function, $A_i$ is the corresponding coefficient of the order $i$, and $r$ is the radial coordinate of HDOE. A more detailed description is given in the Supplementary File.

The diffraction efficiency of HDOE at wavelength $\lambda$ is given by the following formula [24]:

$$\eta_{\text{HDOE}}(\lambda) = \sin c^2 \left( \frac{m\lambda_0}{\lambda} - k \right),\tag{2}$$

where $\lambda_0$ is the central spectral wavelength and $k = 1, 2, 3, 4 \cdots$ is the diffraction order that occurs. The diffraction efficiency curve of the diffractive lens in the visible spectrum is shown in Figure 2a.

The image captured by the detector of SHDCIS has a color cast phenomenon, which is caused by two reasons; one is that the diffraction efficiency of HDOE varies in the broadband spectrum, which means that the ability of HDOE to focus different light waves is different; the other one is that the quantum efficiency of the sensor varies with the light wave. The quantum efficiency of the sensor is the detection efficiency of the sensor to light waves, which is expressed as $\eta_{QE}$. Since the absorption efficiency of the optical signal by

the semiconductor material is wavelength dependent, $\eta_{QE}$ varies with the wavelength. The curve of $\eta_{QE}$ is calibrated and measured by the detector manufacturer, as shown in Figure 2b, where the horizontal axis is the wavelength and the vertical axis is the percentage. The spectral response efficiency to light waves of different wavelengths can be calculated by multiplying the diffraction efficiency of the HDOE by the quantum efficiency of the detector:

$$\eta_{SHDCIS}(\lambda) = \eta_{\text{Harmonic}}(\lambda) \cdot \eta_{QE}(\lambda). \tag{3}$$

Next, the response efficiency for the light passing through the optical system on the three RGB channels of the detector can be calculated:

$$\kappa_{SHDCIS} = \int_{\lambda_1}^{\lambda_2} \eta_{SHDCIS} d\lambda, \tag{4}$$

where $(\lambda_1, \lambda_2)$ is the wavelength response range of one of the sensor's RGB channels.

The calculated response efficiency ratio of the three channels of RGB was 1:1.3324:1.1123, and the contrast shift ratio of the image measured by the experiment was 1:1.2983:1.1006. The calculated value was approximately consistent with the experimentally measured value. The slight deviation may have been due to the HDOE fabrication error or the quantum efficiency measurement error of the sensor.

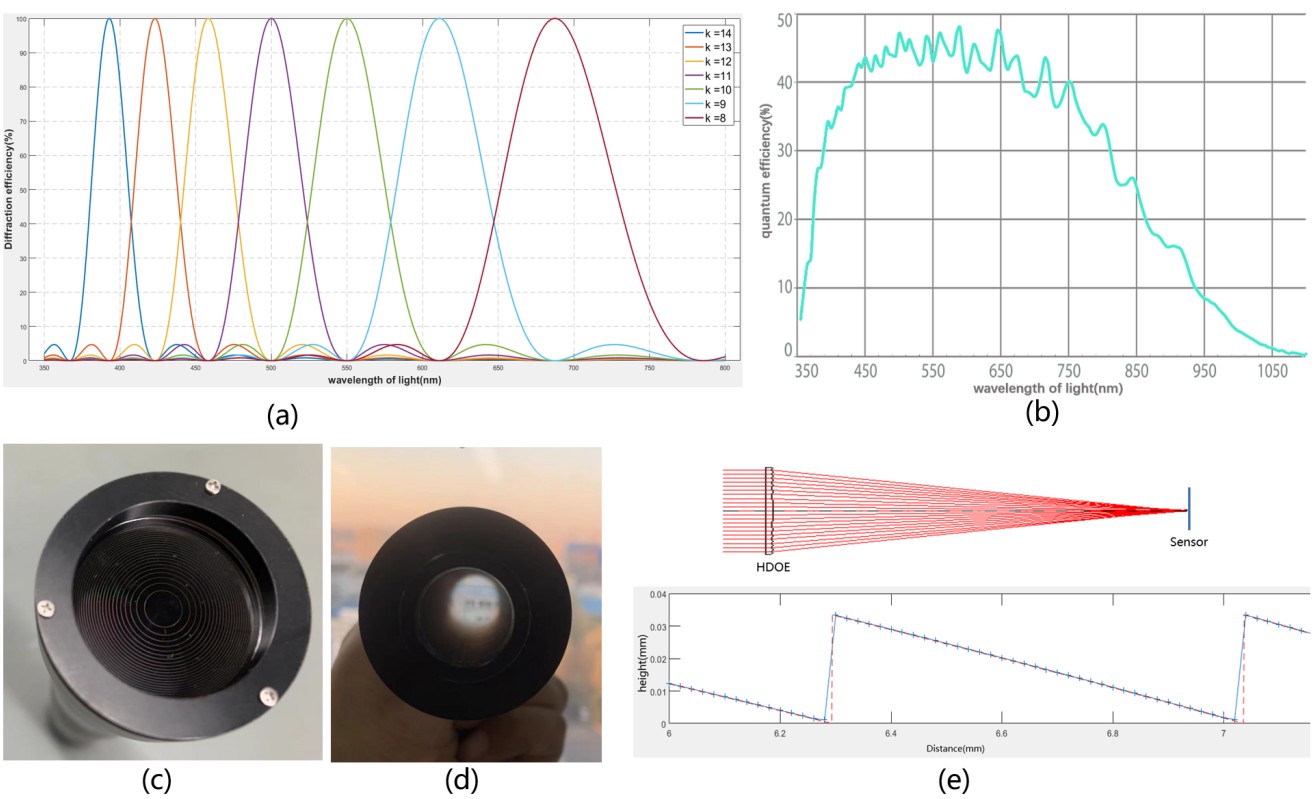

**Figure 2.** (**a**) The diffraction efficiency of the HDOE designed in this study with $\lambda_0$ = 555 nm and $m = 10$ in the full spectrum of visible light; (**b**) the quantum response efficiency curve of the sensor; (**c**) the proposed harmonic diffractive lens with diameter 32 mm and focal length 275 mm that was fabricated from polymethyl methacrylate (PMMA) using diamond single-point machining; (**d**) the built diffraction imaging system consisting of a single harmonic diffraction lens, a lens barrel, and a visible light RGB three-channel detector with 512 × 512 pixels; (**e**) SHDCIS optical structure diagram and HDOE surface structure detection results.

*2.2. Analysis of Image Quality Degradation of HDOE Imaging*

The imaging degradation of HDOE is mainly caused by strong chromatic aberration and decreased diffraction efficiency. The effect of chromatic aberration can be analyzed in the impulse response function of the main order diffraction of the light wave in the HDOE. When the diffraction efficiency of the light wave decreases, a part of the energy will be scattered and dispersed on the image plane, forming stray light background radiation. Then, the effect on the reduction of diffraction efficiency can be analyzed by the impulse response of the light wave diffracted in the non-design order.

According to the Fresnel diffraction theory, a light wave with an amplitude of 1 on the HDOE plane$(x', y')$ passes through the HDOE; the complex amplitude expression of the wave field on the image plane $(x, y)$ with distance $z$ is [8]:

$$U_\lambda(x, y, z) = \frac{e^{ikz}}{i\lambda z} \cdot \iint t_{HDOE,\lambda}(x', y') e^{\frac{ik}{2z}[(x-x')^2 + (y-y')^2]} dx' dy',$$ (5)

where $\lambda$ is the wavelength of light, i is the wave number, and $i = 2\pi/\lambda$, $t_{HDOE,\lambda}$ is the complex amplitude transmittance function of HDOE. The complex amplitude transmittance function is defined as the ratio of the complex amplitude of the outgoing light field to the complex amplitude of the incident light field on the plane of the element, which can express the influence of the element on the incident wavefront. $t_{HDOE,\lambda} = e^{i[\phi_{HDOE,\lambda}(x',y')]}$, $\phi_{HDOE,\lambda}$ is the phase delay function of HDOE, which can be derived from the phase function $\varphi(r)$. For a detailed derivation see the supplementary File.

Thus, the impulse response function of the main order diffraction of the light wave on the image plane is:

$$h_1(x, y) = |U(x, y)|^2$$ (6)

When the diffraction efficiency is lower than 100%, the light waves are diffracted in other non-design orders, and a part of the energy is not concentrated in the focus, resulting in the background radiation of stray light. Consquently, the light field intensity function of the HDOE imaging system can be written as the superposition of the focusing impulse response $h_1(x, y)$ and the background impulse response $h_{BG}$:

$$I(x, y) = |h_1(x, y)|^2 + |h_{BG}(x, y)|^2$$ (7)

The point spread function of HDOE is a normalized representation of the light field intensity function, and the normalization process can be expressed as:

$$h(x, y) = |I(x, y)|_1 = \left| |h_1(x, y)|^2 + |h_{BG}(x, y)|^2 \right|_1$$ (8)

A further simplified approximation can be obtained:

$$h(x, y) = \eta_{HDOE} h_1(x, y) + (1 - \eta_{HDOE}) h_{BG}(x, y),$$ (9)

where $h_{BG}(x', y') = \int_{-\infty}^{\infty} \int_{-\infty}^{\infty} (1 - \eta_1) \delta(x - x', y - y') dx' dy'$, $\delta(\cdot)$ is the unit impulse function. For a detailed derivation, please refer to the Supplementary File.

Thus, the imaging degradation model of HDOE in the bandwidth spectrum can be expressed as:

$$g = [\eta_{HDOE} h_1(x, y) + (1 - \eta_{HDOE}) h_{BG}(x, y)] * f + n$$ (10)

The image quality degradation model includes various influencing factors that affect diffraction imaging, where $(1 - \eta_{HDOE}) h_{BG}(x, y)$ represents the influence of stray light caused by the diffraction of light waves in non-design orders, $\eta_{HDOE}$ represents the influence of diffraction efficiency, and $h_1(x, y)$ represents the influence of chromatic aberration. This model comprehensively and accurately describes the image degradation process of diffraction imaging and provides a useful reference for further image restoration research.

The distribution of the PSF of the SHDCIS on the focal plane was calculated according to the design value, as shown in Figure 3a. The real PSF of the SHDCIS was measured and calculated according to the method provided in [25], as shown in Figure 3c. The figure shows that the PSF of the SHDCIS is basically close to the design value. The PSF has obvious spatial variation, and there is a clear difference in the PSF between the central and off-axis fields. Consequently, off-axis fields of view will show more aberration than central fields of view. This means that the computation and reconstruction of images in the central and off-axis fields of view differ in the intrinsic process of image restoration. This conclusion provided inspiration for the design of the regional division and channel selection module in the DDZMR network.

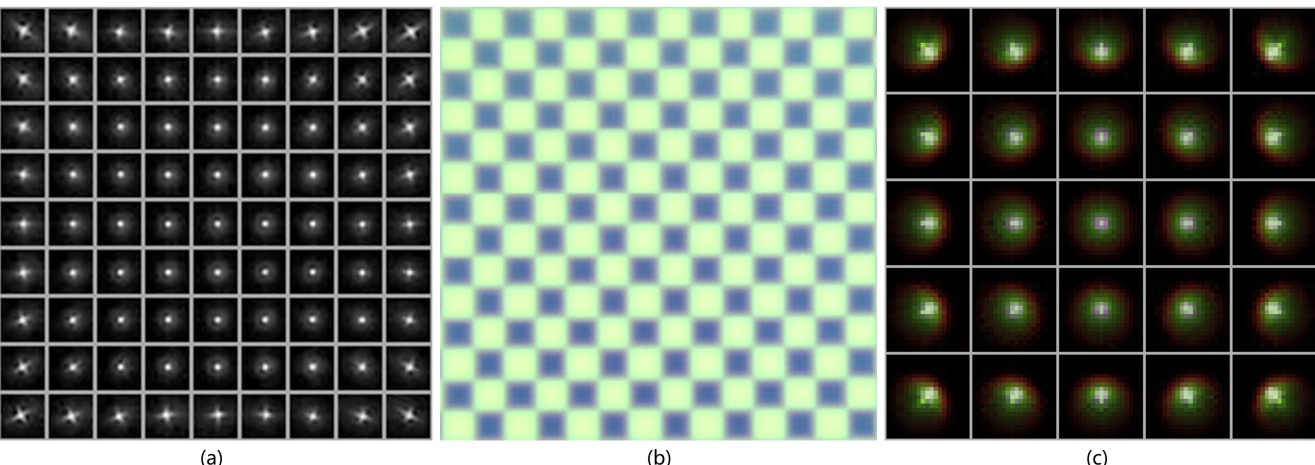

(a)    (b)    (c)

**Figure 3.** (**a**) The PSF distribution of the harmonic diffractive lens with a wavelength of 500–600 nm on the focal plane; (**b**) PSF estimation experiment of harmonic diffraction imaging system; (**c**) The PSF distribution of the imaging system measured experimentally.

## 3. Deep Dense Zoned Multipath Residual Network for Image Restoration and Simultaneous Super-Resolution Image Reconstruction

### 3.1. Deep Dense Zoned Multipath Residual Network

In SHDICS, we built a multi-task network model for image restoration and super-resolution, in order to recover the degradation of diffraction imaging and improve the imaging capability of the system, because data-driven image restoration and super-resolution processing have the same inherent process. The process is to find the optimal intrinsic relationship in the pixel-to-pixel correspondence of the input image with the output image and to remove only the unwanted content, while carefully preserving the desired spatial details. Therefore, we reconceptualized image restoration and super-resolution reconstruction as two tasks performed simultaneously within a process.

Accordingly, the imaging degradation procedure in SHDCI can be expressed as:

$$L = h \otimes (H\downarrow_s) + n, \tag{11}$$

where $L$ is the low quality image captured by the sensor, $H$ is the ground truth, $h$ is the degradation function of HDOE derived in Section 2.2, which can also be called fuzzy kernel, $\downarrow_s$ represents the downsampling process, and $n$ is noise generated by the system.

The residual network framework can build very deep networks to perform multi-image tasks [16,26]. Consequently, based on the residual network and the image quality degradation model of HDOE, we designed the deep dense zoned multipath residual (DDZMR) network. The main components of DDZMR network are: the image regional division and channel selection module; the double residual tandem spatial attention block (DRTSAB); the residual dense concatenation block (RDCB); an upsampling module. These components are combined in an end-to-end architecture for DDZMR learning; the network model is shown in Figure 4.

The image regional division and channel selection module divides the image into 25 $128 \times 128$ pixel subimages. The pixel overlap for adjacent subimages is 32 pixels. Each subimages learns a subregion-specific degradation process along its own independent channel. Since the performance of PSF in HDOE is different on different field of views, the region partitioning strategy can make the DDZMR network better learn the sub-image computational reconstruction process. The subimages maintain the original resolution features along the network hierarchy, thereby minimizing the loss of precise spatial detail. The reconstructed subimages are finally stitched together into a reconstructed full-frame image.

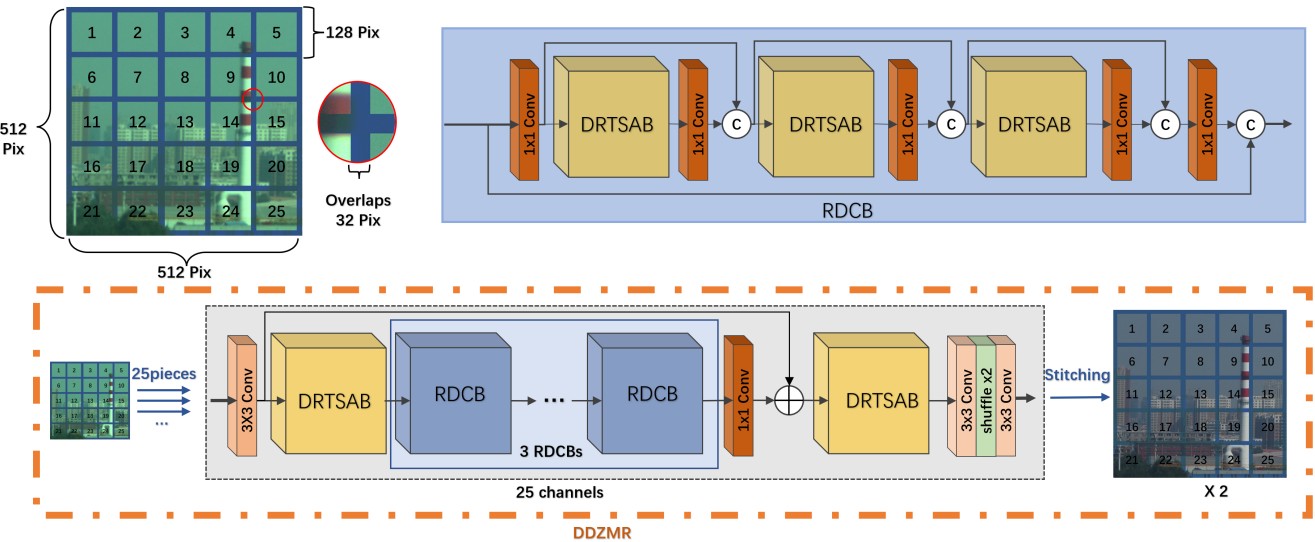

**Figure 4.** Deep dense zoned multipath residual network.

DRTSAB consists of two sets of convolutional (Conv) and spatial attention module (SAM) layers, as shown in Figure 5, and the input and output of each layer are connected to subsequent layers for local residual learning. The SAM is connected to the residual block to filter out blur caused by chromatic aberration and loss of detail caused by stray light, and optimize the network size. The SAM independently applies global average pooling and max pooling operations to feature extraction in the channel dimension, and concatenates the outputs to form feature maps. It aims to exploit the spatial interdependence of convolutional features to generate spatial attention feature maps, which are used to recalibrate the incoming feature space [26].

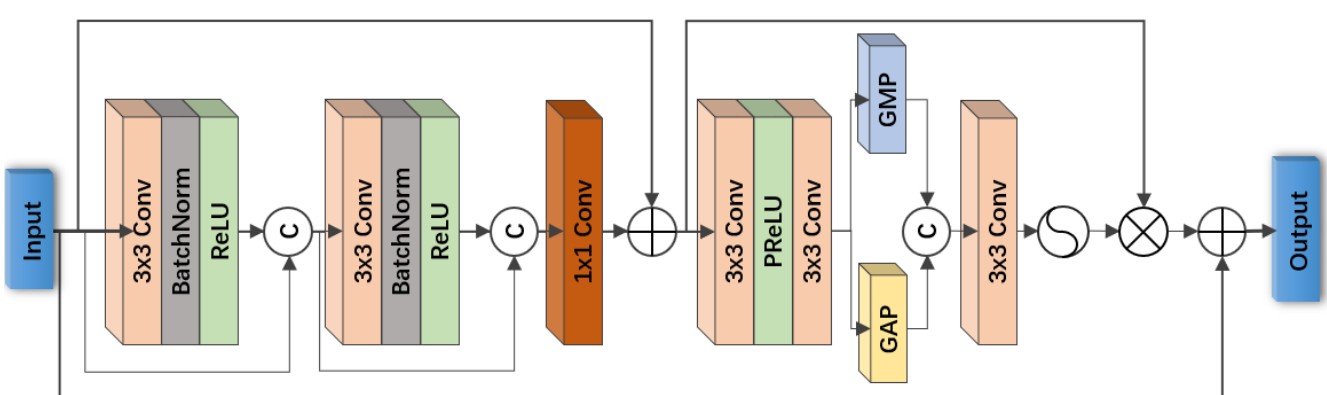

**Figure 5.** Double residual tandem spatial attention block (DRTSAB).

The components of RDCB are three dense skip-connected DRTSABs to propagate low-level features to subsequent residual blocks. The purpose of the dense skip connections is to further facilitate the back propagation of gradients, to reuse low level features to increase the potential of the final reconstructed features [26], and to facilitate information flow during learning. We utilized a $1 \times 1$ Conv layer for each dense skip connection to adaptively modulate low-level features in different layers.

The DDZMR network takes $L$ as input and first uses a $3 \times 3$ Conv layer to capture shallow features; one DRTSAB is used to learn nonlinear high-level features; then, three RDCBs are concatenated to learn to utilize various information from the L image space so that the network can access richer information; finally, the feature maps are input to DRTSAB for final feature selection and filtering. Residual blocks have been used in various state-of-the-art architectures [18,20,26–28], as they have been shown to improve training. Residual learning is applied to each block to facilitate information flow through skip connection paths.

In the upsampling module, the subpixel convolution strategy proposed in [28] was used. This consisted of a $3 \times 3$ Conv layer and a Shuffle layer to rearrange the data in the channels to different spatial locations to enlarge the features. Finally, the upsampled feature maps are sent to convolutional layers to reconstruct the final three-channel RGB image.

### 3.2. Multimodal Loss Function

According to the analysis in Section 2.2, it can be concluded that the degradation of HDOE imaging is a complex process, including blurring and edge artifacts caused by chromatic aberration, and stray light pollution caused by non-design order diffraction of light waves. Consquently, four loss function components are tailored to supervise learning various aspects of HDOE image degradation.

The learning function is $G : L \rightarrow H$, where $L$ is the low-quality image captured by the detector and $H$ is the ground truth. Therefore, this process can be expressed as $\hat{H} = G(L)$, where $\hat{H}$ is the high-quality image generated by the DDZMR. The loss function of the DDZMR network is formulated as follows:

$$L_{DDZMR} = \lambda_i L_{\text{information}} + \lambda_{c1} L_{color} + \lambda_{c2} L_{\text{chromatic}} + \lambda_s L_{stray-light} \tag{12}$$

The image information and detail loss function was composed of the standard cross entropy function [29], which quantifies the difference in pixel intensity distribution between the reconstructed image $\hat{H}$ and the ground truth $H$. For a total of $N_H$ pixels in $\hat{H}$, $L_{\text{information}}$ is calculated as:

$$L_{\text{information}} = \frac{1}{N_H} \sum_{i=1}^{N_H} [-H_i log \hat{H}_i - (1 - H_i) log(1 - \hat{H}_i)] \tag{13}$$

The contrast evaluation loss function evaluates and calculates the overall similarity of the real measurement values in the RGB space of the reconstruction $\hat{H}$ and $H$, respectively [30]. The images captured by SHDCIS have contrast differences between the three RGB channels and the real image. This difference is quantified by expressing $\Delta r$, $\Delta g$, and $\Delta b$ as the per-channel numerical difference between $\hat{H}$ and $L$. Thus, the contrast evaluation loss function is expressed as:

$$L_{\text{color}} = \left\| 4(\Delta r - \Delta g)^2 + (\Delta r + \Delta g - 2\Delta b)^2 \right\|_2 \tag{14}$$

The chromatic aberration loss function designed to evaluate artifacts and blurring at the edges of images, which are mainly caused by the chromatic aberration of HDOE. Local image gradients get used to measure the effect of chromatic aberration correction in DDZMR network output images. The standard $3 \times 3$ Sobel operator to calculate the spatial

gradient of image $I$ is expressed as: $\nabla I = \sqrt{I_x^2 + I_y^2}$ [31]. Thus, the chromatic aberration loss functions are defined as [32]:

$$L_{chromatic} = \left\| |\nabla H|^2 - |\nabla \hat{H}|^2 \right\|_1 \tag{15}$$

The purpose of the stray light loss function is to force the image generated by the network to restore the feature content similar to the ground truth, and remove the haze-like noise pollution, which is caused by the stray light pollution formed by the non-design order diffraction of the light wave on the HDOE. The haze-like noise pollution measurement function is defined as the high-level features extracted by the last convolutional layer of the pre-trained VGG-16 network [33]. Consequently, the stray light loss function is expressed as:

$$L_{stray-light} = \left\| \Phi_{VGG}(H) - \Phi_{VGG}(\hat{H}) \right\|_2 \tag{16}$$

## 4. Imaging Experiment and Analysis

### 4.1. Network Training Details

The computational imaging processing module is deployed on a workstation computer equipped with an Intel i7-1070k CPU at 5.1 GHz, 32 GB memory, an Nvidia GeForce RTX 3060 Ultra W 12G, operating on Windows 10. The training set of images used for network training was acquired in a laboratory darkroom, following the method described in [4]; high-resolution real images H were projected onto the screen; and images were captured by SHDCIS to obtain low-resolution degraded images. The training dataset contained 1800 images. The DDZMR network is built based on Python3.8, PyTorch1.9.0, and CUDA10.2. The network optimizer used the Adam algorithm, with $\beta_1 = 0.9$, and the initial learning rate was set to $1 \times 10^{-14}$. Furthermore, the learning rate was then halved every 10 k iterations. It took about 45 h to train the network and about 431 milliseconds (ms) to reconstruct an image using DDZMR. We have accelerated training using a CUDA-linked graphics processing unit (GPU). It is necessary to use a GPU in deep learning networks to improve the operation speed, which can improve the learning efficiency of the network in massive data.

### 4.2. Imaging and Computational Reconstruction

The SHDCIS outdoor imaging test is shown in Figure 6. The image captured directly by the sensor has strong blur, artifacts, noise, and chromaticity differences, as we analyzed, shown in Figure 6a,d. After the calculation and reconstruction of the DDZMR network, the degradation of the optical system imaging had been effectively recovered, and the image quality had been restored, as shown in Figure 6c,f. Furthermore, this was compared with the computational reconstruction algorithm proposed in [10], as shown in Figure 6c,f. It can be easily judged by the human eye from the enlarged detail images that the DDZMR network performs better in detail recovery and removal of blur and artifacts. Therefore, the experiments show that the DDZMR network model successfully replaces the secondary mirror group responsible for complex aberration correction; the SHDCIS imaging performance is satisfactory; and the DDZME network can additionally improve the system capability for image super-resolution.

In order to verify the super-resolution performance of SHDCIS, we carried out a photographic experiment on the resolution test target table in the darkroom laboratory. As shown in Figure 7, after the reconstruction of the DDZER network, the resolution target has become very clear, and the smallest target that can be clearly photographed has increased from the *No. 5* target in the fourth group to the *No. 1* target in the fifth group. After calculation, it can be seen that the resolution of the system is increased to 1.412 times of the original after the calculation and imaging module. This inspiring capability improves the performance of imaging systems with unchanged hardware, which is worth extending to other imaging systems.

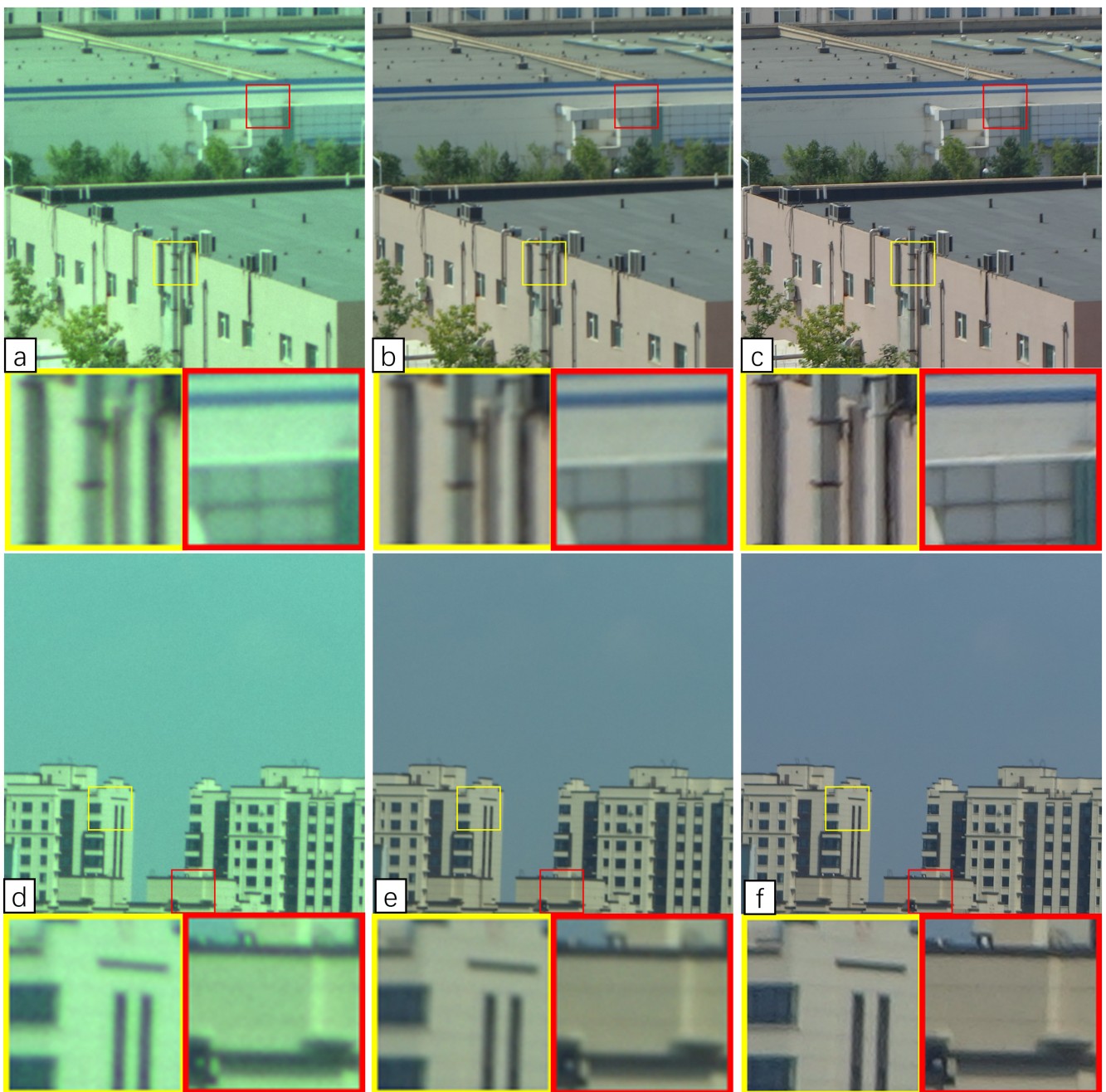

**Figure 6.** SHDCIS outdoor imaging experiment; (**a**,**d**) are the raw images captured by the detectors taken outdoors by SHDCIS; (**b**,**e**) are the processed results using the convolutional neural network proposed in [10]; (**c**,**f**) are the reconstructed images calculated by the DDZMR network model.

### 4.3. Ablation Study Evaluation

We conducted ablation experiments to verify the effectiveness of the multimodal loss function and the image regional division and channel selection module. The multimodal loss function was removed, making the loss function $L_{DDZMR} = L_{information}$. The network was then trained with the same parameters, and the effectiveness of the trained network was tested, as shown in Figure 8, the chroma-specific contrast of the image was not recovered, and the blur and haze of the image were not effectively removed. Next, the regional division and channel selection module was removed, and the same steps were used to test the configuration. As shown in Figure 9, image restoration was minimal, and the edge artifacts of the image were more noticeable than before processing.

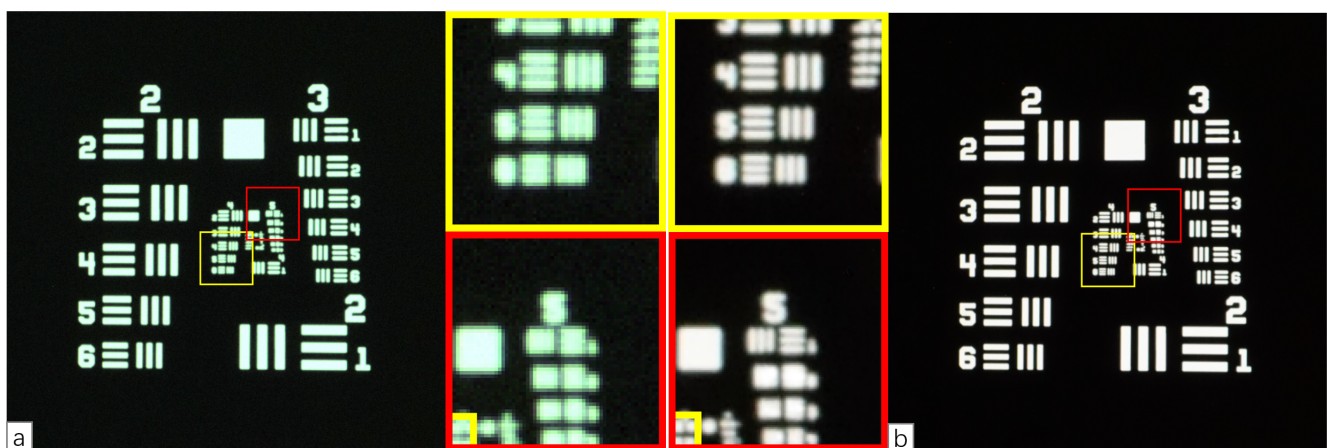

**Figure 7.** SHDCIS imaging test of resolution target; (**a**) is the image captured directly by the detector; (**b**) is the image reconstructed by the DDZMR network.

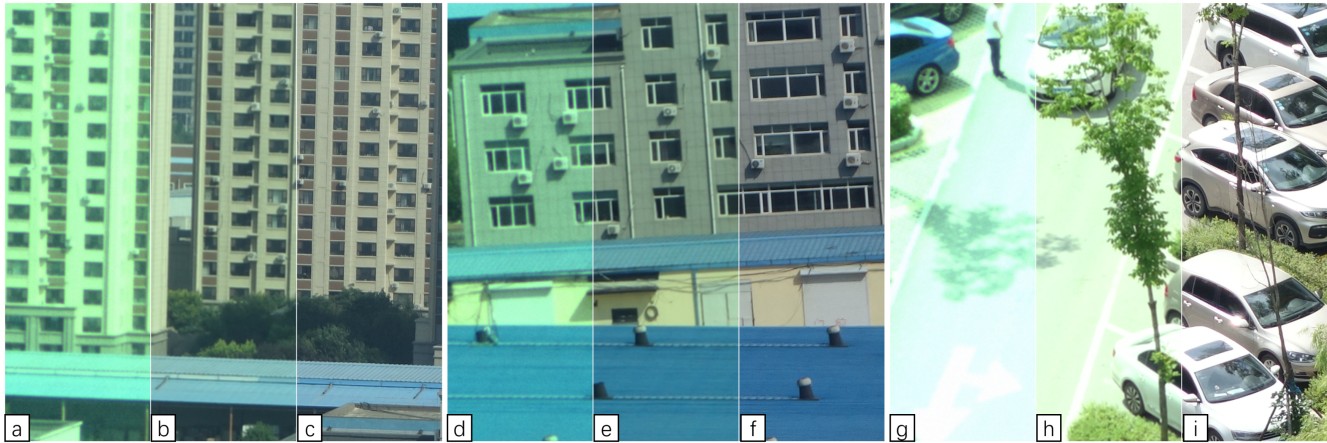

**Figure 8.** (**a**,**d**,**g**) are the original images captured by the sensor; (**b**,**e**,**h**) are the images after processing when the multimodal objective function was replaced by the standard cross-entropy function; (**c**,**f**,**i**) are the results of normal processing by the DDZMR network model.

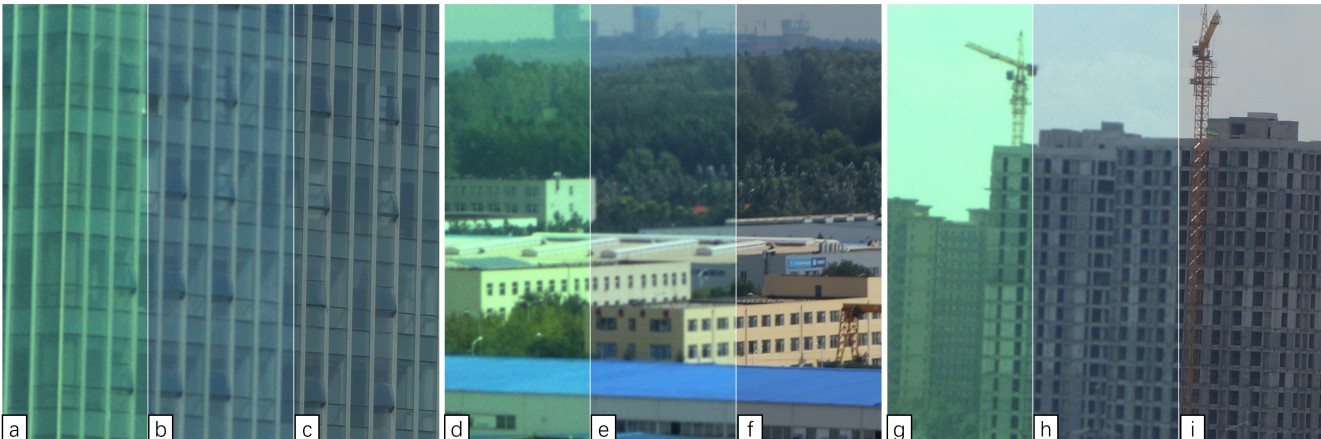

**Figure 9.** (**a**,**d**,**g**) are the original images captured by the detector; (**b**,**e**,**h**) are the paths of 25 regions replaced with a 1-block area; (**c**,**f**,**i**) are the results of normal processing by the DDZMR network model.

A quantitative comparison of the Ablation testing is shown in Table 1. Peak Signal to Noise Ratio (PSNR) and Structural Similarity (SSIM) enable quantitative objective evaluation of reconstructed images. PSNR mainly evaluates the ratio of the maximum possible

power of the real signal of the image to the destructive noise power that affects its representation accuracy. SSIM evaluates the quality of the image from three aspects: Luminance, Contrast, and Structure. The PSNR and SSIM of images processed by the DDZMR network are improved by 12% and 4% compared to the ablation component, and the training time of the network is shortened by 33%. Consequently, experiments show that the multi-modal loss function customized according to the HDOE imaging image quality degradation model is effective for SHDCIS, which can make the DDZMR network restore the image quality better. When the degree of image degradation in the image plane space under different off-axis fields of view is inconsistent, the image regional division and channel selection module can help the network to better learn the degradation process of images at different positions, and improve the convergence speed of the network.

**Table 1.** Quantitative comparison of ablation multi-modal loss function and area division and channel planning modules.

| Method | PSNR | SSIM | Training Time (h) | Processing Time (ms) |
|---|---|---|---|---|
| DDZMR | 28.69 | 0.7045 | 45 | 413 |
| DDZMR without the multimodal loss function | 25.57 | 0.6754 | 53 | 402 |
| DDZMR without the image regional division and channel selection module | 26.78 | 0.6842 | 67 | 425 |

### 4.4. Application of the DDZME Network in Normal Image Super-Resolution Tasks

To evaluate the performance and robustness of the DDZMR network for image super-resolution, we tested it with the publicly available benchmark datasets B100 [34] and Urban100 [35] data, and compared it with the state-of-the-art networks. B100 [34] contains various animals, plants, and other natural scenes, and Urban100 [35] contains only scenes of urban buildings and roads. We quantified the performance of each dataset on a common test set; according to [20], PSNR and SSIM quantify the reconstruction quality and structural similarity of the generated images relative to the ground truth. Some qualitative comparisons are shown in Figure 10. Bicubic often suffers from oversaturation and VDSR [20] does not perform well in tonal correction. In comparison, LapSRN [21], CARN [18], and DDZMR were generally better in terms of color restoration, contrast, and image detail enhancement after super-resolution. The images created by DDZMR network, in addition to achieving comparable color recovery and tone correction, were also reasonably clear and the network performed excellently.

A quantitative comparison of PSNR and SSIM for each network image super-resolution performance is shown in Table 2. As with the intuitive judgment of images, VDSR [20] has the lowest PSNR and SSIM values, LapSRN [21] and CARN [18] perform well, and DDMZR networks are also competitive, generally performing better in PSNR and SSIM.

**Table 2.** Quantitative comparison of existing methods on the B100 [34] and Urban100 [35] datasets.

| Method | B100:PSNR | B100:SSIM | Urban100:PSNR | Urban100:SSIM |
|---|---|---|---|---|
| Bicubic | 27.86 | 0.6909 | 26.41 | 0.6195 |
| VDSR [20] | 31.52 | 0.8873 | 29.65 | 0.9048 |
| LapSRN [21] | 31.69 | 0.8929 | 30.40 | 0.9107 |
| CARN [18] | 32.04 | 0.8963 | 31.72 | 0.9234 |
| **DDZMR (Ours)** | **32.09** | **0.8975** | **31.82** | **0.9247** |

Although DDZMR has only a little performance improvement in super-resolution processing compared to CARN, DDZMR has the ability to simultaneously restore degraded images, which is a capability that CARN and other networks do not have. The image in HDOE imaging is severely degraded. Because CARN cannot effectively restore degraded

images, it is very ineffective in super-resolution processing of images captured by SHDCIS. See the supplementary File for specific image processing details.

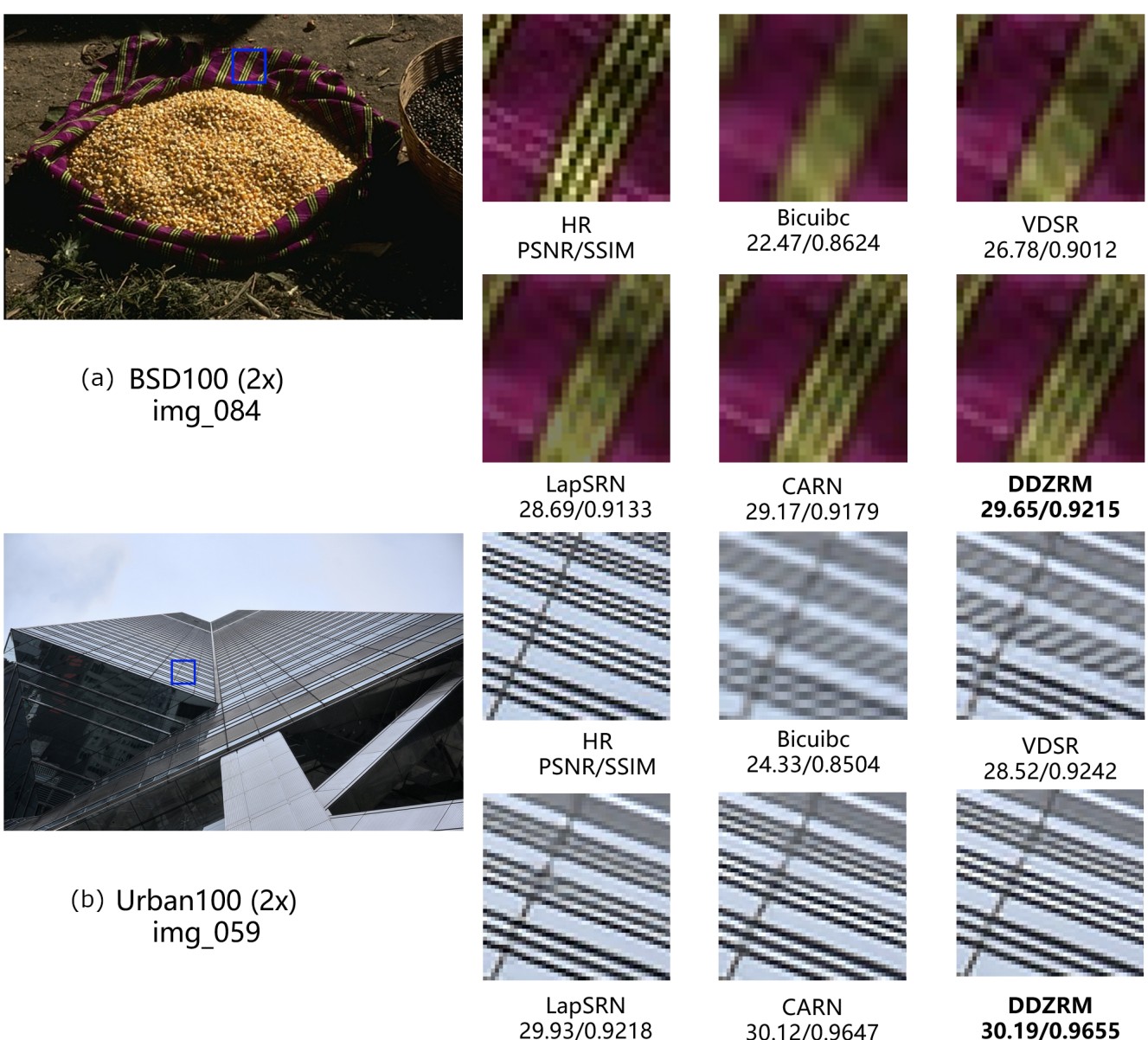

**Figure 10.** (**a**) Comparison of the performance of DDZMR networks and the state-of-the-art networks on dataset B100 [34]; (**b**) Comparison of the performance of DDZMR networks and the state-of-the-art networks on dataset Urban100 [35].

## 5. Test Results

In the experiments in this work, the performance of the DDZMR network in computationally reconstructing the degraded images of HDOE is first tested based on the prototype SHDCIS. Compared with the cross-channel convolutional neural network proposed in [10], DDZMR performs better on degraded diffraction images, and it can also perform $2\times$ image super-resolution simultaneously. Then, the enhancement effect of DDZMR network on SHDCIS imaging resolution was tested in the dark room, which showed that DDZMR could improve the resolution by 1.412 times. Next, an ablation study is performed on the DDZMR component, which verifies the effectiveness of the image regional division and channel selection module and multimodal loss function. Finally, we test the robustness of the ISR capability of the DDZMR network using publicly available datasets and compare it

with the SOTA method; DDZMR shows excellent performance, and it has image restoration performance that other ISR networks do not have.

## 6. Conclusions and Discussion

The imaging degradation of DOE in the visible spectral bandwidth caused by chromatic aberration and reduced diffraction efficiency is a long-standing problem, which limits the development and application of DOE imaging systems [36]. The application of computational imaging enables single-lens diffractive lenses to achieve high-quality imaging in the visible light bandwidth [37]. It is an essential step to analyze the image quality degradation of diffraction imaging with a wide-bandwith spectrum and to design a suitable high-performance computing imaging algorithm. In this paper, we have analyzed the image quality degradation model of HDOE imaging, and based on the analysis, we designed a DDZMR network model that performs image restoration and image super-resolution reconstruction simultaneously as the computational imaging component of SHDCIS. The SHDCIS physical prototype achieved high-quality imaging, and the ISR effect was able to improve 1.412 times. Furthermore, the DDZMR network showed excellent robustness in non-specific super-resolution tasks and denoising tasks.

Therefore, the DDZMR network can also be applied in other imaging fields, such as microscopic imaging. In microscopic imaging, the object distance of an objective is no longer considered infinite, but rather a very short distance. Subtle objects produce an inverted, magnified image close to the focal point of the objective. Accordingly, the image degradation process for microscopic imaging is different compared to that of ordinary photographic lenses. Adjusting the loss function according to its degenerate model characteristics can make the DDZMR network get better performance. It is worth mentioning that the region division and channel planning modules that perform multi-path partitioning will play a great role in performing image restoration tasks, where the blur kernel changes greatly with position changes. This module can be generalized to more image restoration tasks, such as out-of-focus image restoration tasks, computational imaging tasks for imperfect optical systems, and deblurring tasks. With appropriate changes to the DDZMR network, it can also handle grayscale images, since there is no cross-channel specific operation for the three RGB channels involved in the DDZMR network framework.

The subsequent research plan is to study the effect of complex environmental conditions (such as changes in temperature and air pressure) on diffraction imaging. Research on computational imaging algorithms to adapt to more complex degraded images, so that SHDCIS can still have good performance under complex environmental conditions.

**Supplementary Materials:** The following supporting information can be downloaded at: https://www.mdpi.com/article/10.3390/app12094753/s1.

**Author Contributions:** Conceptualization, Y.X. (Yongsen Xu) and K.L.; methodology, Y.X. (Yulei Xu); software, K.L.; validation, Y.W., N.D. and Y.Y.; investigation, H.S.; writing—original draft preparation, K.L.; writing—review and editing, X.Y.; funding acquisition, H.W. All authors have read and agreed to the published version of the manuscript.

**Funding:** National Defense Science and Technology Innovation Fund of the Chinese Academy of Sciences (CXJJ-19S014); The National Key Research and Development Program of China.

**Institutional Review Board Statement:** Not applicable.

**Informed Consent Statement:** Not applicable.

**Conflicts of Interest:** The authors declare no conflict of interest.

## Abbreviations

The following abbreviations are used in this manuscript:

| | |
|---|---|
| DOE | Diffractive optical elements |
| HDOE | Harmonic diffractive optical element |
| MDL | Multi-order diffractive lens |
| DDZMR | Deep dense zoned multipath residual |
| SHDCIS | Single-lens harmonic diffraction computational imaging system |
| PSF | Point spread function |
| OTF | Optical transfer function |
| DRTSAB | Double residual tandem spatial attention block |
| RDCB | Residual dense concatenation block |

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
