# Peer review of "Computational Imaging for Simultaneous Image Restoration and Super-Resolution Image Reconstruction of Single-Lens Diffractive Optical System"

_applsci, doi:10.3390/app12094753_

Round 1
Reviewer 1 Report
In this interesting manuscript, Liu et al. introduce a computational imaging module for the restoration of images acquired using single-lens systems. They initially analyzed the spectral properties and image degradation of an HDOE system. Based on these results, they designed a computational network for image restoration and resolution improvement. The results are sound to me. The approach may have the potential for interdisciplinary applications.
I have a few comments:
(1) This work is focused on single-lens systems; however, (chromatic) aberration can also be introduced into images acquired using optical imaging (microscopy), also when corrected objectives are applied.
// Do the authors think their image reconstruction may be applied with modifications (adaption of the image degradation model) to microscopic images? What kind of information would be required to adapt their model? The interdisciplinary application of this approach would be beneficial for scientists.
Please comment on that and include an outlook in the discussion sections (putative applications in other fields).
(2) To my knowledge, ´chromatic aberration´ as stated in the text, is not based on diffraction but on color dispersion (refraction phenomenon when light enters a medium of higher/lower refractive index; e.g., air – lens); on the other hand, diffraction of light at the lens aperture depends on the wavelength. In the text, it is partly unclear which of these phenomena are addressed.
// Please clarify this in the text.
(3) ´Super-resolution` in the imaging field is dedicated to specialized techniques capable of imaging single emitters and reduction of diffraction patterns interfering with the resolution of image objects.
// If indicated, the authors may replace `super-resolution` with e.g., extended-resolution.
(4) Network training took 45h.
// Is it possible to accelerate training using CUDA, i.e., could one adapt the image processing module to access related graphic cards? Would this make sense to extend possible applications of this approach?
(5) Links to table 1 and table 2 are missing in the text; the tables do not show sufficient description; e.g., PSNR, SSIM explanation of these abbreviations are missing.
(6) Although I guess it is clear for the authors, a brief description of the basis of the implementation of the (software) algorithms would be appreciated (used platform, etc.). This would help to use the approach in other scientific fields.
(7) Line 57: Delete question mark next to references.
(8) Line 63: Reference missing.
(9) Although I am not a native speaker, the word order should be revised. Partly it is challenging to understand the meaning of the sentences.
Reviewer 2 Report
The paper raises very important and interesting problem connected with a loss of quality images in case of using e.g. a lens. Very often a structure of images become unclear and illegible. A correction method has to apply to improve the images.
Authors propose a DDZMR network for a single-lens diffractive lens computational imaging system for their tests. Such model can also perform image restoration and its reconstruction.
Authors conducted discussion of obtained results. They shown good and worse sides of the DDZMR network. But I propose to divide a chapter 5. Results and discussion into two chapters: 5. Test results and 6. Conclusions. In my opinion scientific paper should include the last chapter “Conclusions”. Now the paper gives the impression of infinite.
Obtained results are satisfactory but the tests have to be continue. It is connected with comparison between CARN and DDZMR (Table 2). We can see that an advantage of DDZMR is not meaningful.
All used abbreviation in the text should be explained. I know that Authors put a section Abbreviations, but in my opinion my proposal makes easier a reading the paper. Additionally Authors aren’t consistent. The abbreviation DOE is explains in an Abstract, HDOE in an Introduction, but others already not.
Lines 57 ang 63 – sign “?” is used in brackets [], why?
How the method is effective in case of black and white images?
The DDZMR is effective for fragments of images which are magnified. What a magnification scale up was used for conducted tests?
Did Authors calculate a threshold of the magnification for which their method ceases to be effective?
